# Biomimetic catalytic transformation of toxic α-oxoaldehydes to high-value chiral α-hydroxythioesters using artificial glyoxalase I

Sang Yeon Park[1], In-Soo Hwang[1], Hyun-Ju Lee[1] & Choong Eui Song[1]

Glyoxalase I plays a critical role in the enzymatic defence against glycation by catalysing the isomerization of hemithioacetal, formed spontaneously from cytotoxic α-oxoaldehydes and glutathione, to (S)-α-hydroxyacylglutathione derivatives. Upon the hydrolysis of the thioesters catalysed by glyoxalase II, inert (S)-α-hydroxy acids, that is, lactic acid, are then produced. Herein, we demonstrate highly enantioselective glyoxalase I mimic catalytic isomerization of in-situ-generated hemithioacetals, providing facile access to both enantiomers of α-hydroxy thioesters. Owing to the flexibility of thioesters, a family of optically pure α-hydroxyamides, which are highly important drug candidates in the pharmaceutical industry, were prepared without any coupling reagents. Similar to real enzymes, the enforced proximity of the catalyst and substrates by the chiral cage in situ formed by the incorporation of potassium salt can enhance the reactivity and efficiently transfer the stereochemical information.

[1] Department of Chemistry, Sungkyunkwan University, 2066, Seobu-ro, Jangan-gu, Suwon, Gyeonggi 440-746, Korea. Correspondence and requests for materials should be addressed to C.E.S. (email: s1673@skku.edu).

Nature has evolved a wealth of proteins called enzymes that catalyse the chemical reactions necessary to sustain all life on Earth. Glyoxalases (I and II) and glutathione constitute a set of glyoxalase enzymes, which carry out the detoxification of methylglyoxal and other reactive α-keto aldehydes by the sequential action of two thiol-dependent enzymes. First, transition metal containing glyoxalase I catalyses the formation and isomerization of the hemithioacetal, which was formed spontaneously between glutathione (GSH) and α-keto aldehydes, into (S)-α-hydroxyacylglutathione. Next, glyoxalase II hydrolyses these α-hydroxy thioesters, producing (S)-α-hydroxy acid derivatives (for example, (S)-lactate and GSH from (S)-lactoyl-GSH)[1]. Glyoxalase thereby protects cells from α-oxoaldehyde-mediated formation of advanced glycation[2]. In particular, the glyoxalase I-catalysed isomerization of hemithioacetals involves the sequential formation of α-hemithioacetal, deprotonation to form the enediol intermediate and its enantioselective reprotonation[3], leading to the formation of enantiopure α-hydroxythioesters[4–6] (Fig. 1a). From the synthetic chemistry point of view, the glyoxalase I-catalysed detoxification process, that is, conversion of glyoxals with thiol into α-hydroxy thioesters, can be regarded as a variant of enantioselective intramolecular Cannizzaro reactions[7–11].

α-Hydroxythioesters are ubiquitous intermediates for functional group manipulations and C–C bond-forming reactions as illustrated in recent synthetic efforts towards α-hydroxyesters, -acids, -thioacids, -ketones, -aldehydes and -amides[12–17]. Enantiomerically pure α-hydroxy thioesters can be used as chiral synthons in asymmetric synthesis of biologically active natural and unnatural compounds[18–20]. Furthermore, α-hydroxy thioesters by themselves also appear to be vital for diverse pharmaceutical activities including antitumor, anticholinergic, mydriatic and spasmolytic activity[21,22]. Despite advantageous utility of α-hydroxy thioesters, the lack of a general and straightforward catalytic protocol to access enantioenriched α-hydroxy thioesters has hindered their applications in pharmaceutical industry. Thus, developing a general catalytic route to access both enantiomers of α-hydroxy thioesters is highly desirable.

A significant amount of new scientific insight, discovery, and application taking place at the interface of chemistry and biology is related to biomimetic catalysis[23–25]. Therefore, developing powerful biomimetic catalytic system is highly interesting from the perspectives of both biology and chemistry. Given our previous studies on our evolved cation-binding catalyst 1 (refs 26–30), we presumed that catalyst 1 could act as a synthetic glyoxalase I for the enantioselective isomerization of the spontaneously formed hemithioacetal adduct between thiols and α-oxoaldehydes into enantio-enriched α-hydroxythioesters. The hydrogen-bond chelating interaction between the acidic phenolic protons of the catalyst and two oxygen atoms of hemithioacetals would enhance the electrophilicity of the carbonyl carbon atom, consequently increasing the acidity of the α-proton. The soluble fluoride anion generated in situ upon the activation of potassium fluoride (KF) by the chiral cation-binding catalyst can further deprotonate the α-proton of

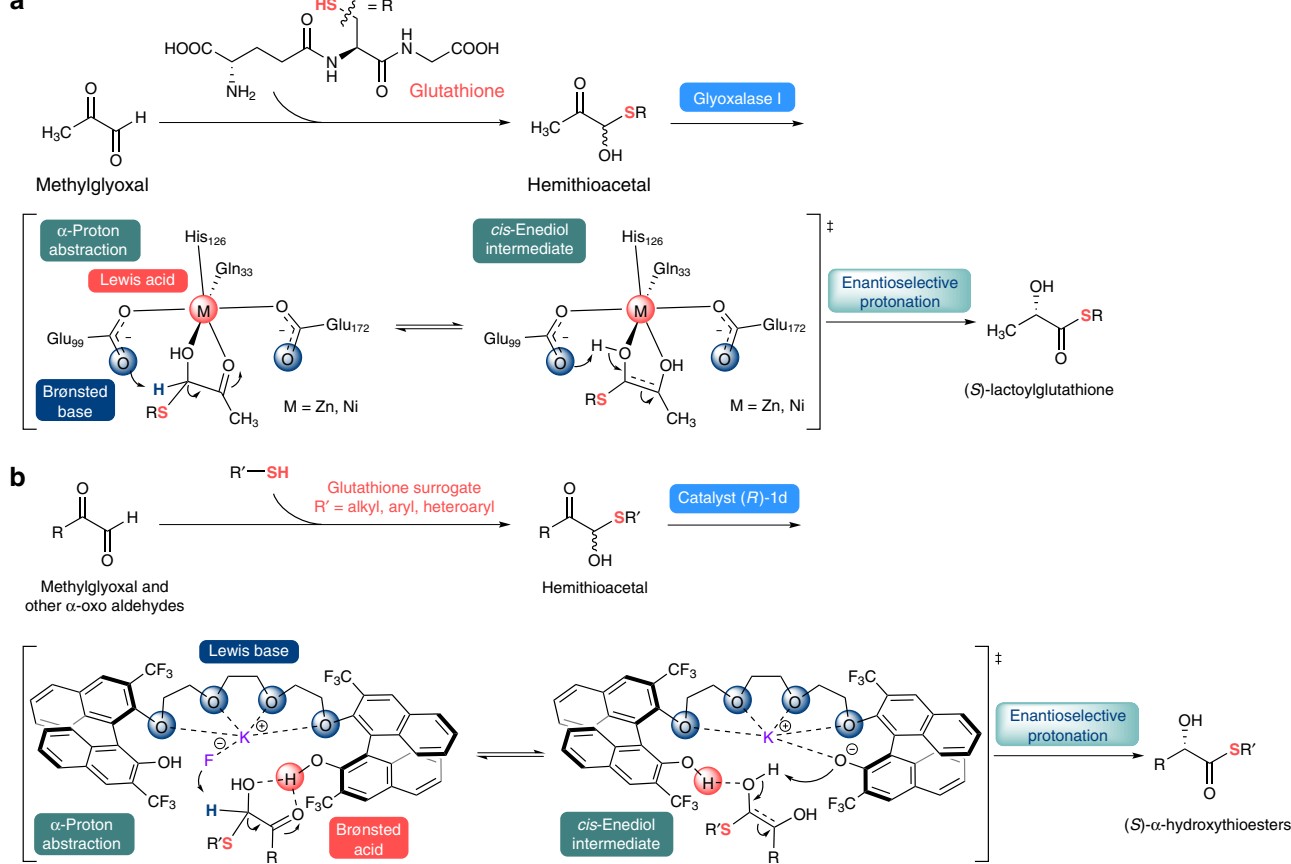

**Figure 1 | Glyoxalase I pathway and glyoxalase I-mimic reaction design.** (**a**) Detoxification of methylglyoxal via enantioselective isomerization catalysed by glyoxalase I. M = Zn for the eukaryotic glyoxalase I such as human and yeast; M = Ni for the prokaryotic glyoxalase I such as the bacteria *Escherichia coli* and *Pseudomonas aeruginosa*. (**b**) Plausible working hypothesis for the enantioselective isomerization of hemithioacetals catalysed by artificial glyoxalase I.

hemithioacetals, producing ene-diol intermediates which can then be stereoselectively reprotonated. In particular, the chiral cage *in situ* formed by the incorporation of potassium salt creates an ideal active site architecture in which reactive substrates are brought into proximity, consequently efficiently transferring the chiral information in the stereo-determining protonation step (Fig. 1b). Of note, on contrast to enzymes, enantioselective introduction of a proton to transient enediol intermediate via synthetic route might be extremely challenging to control it in terms of enantioselectivity due to the small size of the proton[3,7–11].

Here we report an artificial glyoxalase I, which successfully catalyses the enantioselective isomerization of hemithioacetal adducts, spontaneously formed between a range of toxic 2-oxoaldehydes and thiols as GSH surrogates, into high-valued chiral α-hydroxy thioesters with excellent yields and enantio-selectivities. To the best of our knowledge, this is the first successful example of the biomimetic catalytic asymmetric synthesis of α-hydroxy thioesters. Very few examples of biomimetic isomerization of hemithioacetals have been reported to date[8,31,32]. The enforced proximity of the catalyst and substrates by a chiral cage *in situ* formed by the incorporation of potassium salt can enhance the reactivity and efficiently transfer the stereochemical information, mimicking the action of enzymes.

## Results

### Catalyst screening and optimization of the reaction conditions.
To confirm our assumption, phenyl glyoxal hydrate (2a·H₂O) and benzyl thiol (3a) were chosen as model substrates. In the presence of catalyst **1** (10 mol%) and KF (2.0 equiv.) as a base, the effect of the catalyst structure ((*R*)-**1a**–**1c**) on the reaction outcome was first investigated in *o*-xylene (entries 1–3, Table 1). As per our expectations, the hemithioacetal **4a** was formed as a racemic form almost spontaneously under the reaction conditions and its enantioselective isomerization proceeded smoothly, affording the desired enantioenriched **5a**. Based on our knowledge of the catalytic performance of chiral oligoEGs **1** (refs 26–30), the presence of the polyether backbone, the ether chain length (entries 6 and 7) and the acidity of phenolic protons (entry 2 versus entry 4) are crucial in achieving the catalytic activity and enantioselectivity. Although (*R*)-**1c** showed distinguished catalytic activity and enantioselectivity (see, entry 3) compared to (*R*)-**1a** and (*R*)-**1b** (entries 1 and 2, respectively), the observed activity and enantioselectivity were still unsatisfactory (68% conversion and 87% ee after 48 h). Much to our delight, however, the introduction of the electron-withdrawing substituent CF₃ (catalyst (*R*)-**1d**) and C₂F₅ (catalyst (*R*)-**1e**), rather than the iodo-substituent of (*R*)-**1c** at the 3,3′-position of the binaphthyl scaffold, resulted in a nearly complete conversion after the same reaction time, with excellent enantioselectivities (98% and 94% ee in entries 4 and 6, respectively). Notably, the isomerization reaction of the isolated hemithioacetal **4a** using (*R*)-**1d** as the catalyst showed also excellent yield and enanantioselectivity (see, entry 5). The slightly lower enantioselectivity of (*R*)-**1e** compared with that obtained with (*R*)-**1d** can be attributed to the sterically more demanding size of the C₂F₅ substituent. A total loss in catalytic activity was observed when phenol groups were replaced by OMe ((*R*)-**1h**) (entry 9), further confirming the importance of an acidic phenol moiety for catalysis. Full conversion was also achieved with the catalysts (*R*)-**1f** and (*R*)-**1g**, affording the racemic product (entries 7 and 8). This result shows that the size of chiral cage is critical for inducing enantioselectivity. As shown by the catalytic result obtained with catalyst (*R*)-**1i** in which the polyether chain was replaced with an alkyl chain, the polyether backbone is crucial in achieving the observed catalytic activity (entry 10). In further experiments, different solvents were examined with the catalyst (*R*)-**1d** as the optimal catalyst (entries 11–17). Although a range of aprotic nonpolar solvents such as toluene, *o*-, *m*- and *p*-xylene and mesitylene afforded very high enantioselectivity (entries 4 and 11–14), *o*-xylene (entry 4) proved best with respect to both chemical yield and enantioselectivity. However, polar solvents such as CH₂Cl₂, THF and acetonitrile proved to be worse in terms of yields and asymmetric induction (entries 15–17, Table 1).

**Substrate scope of the reaction**. Next, we examined the effect of the structure of thiols as GSH surrogates on the reaction rate and enantioselectivity. Regardless of the steric and electronic nature of the aromatic substituents, all benzyl-type thiols examined in this study gave the desired products **5ab**–**5af** in excellent yields and with excellent enantioselectivities (97–99% yields, 93–99% ee). Primary, secondary and tertiary alkyl thiols were also found to serve as suitable GSH surrogates (see **5ag**–**5ai** in Fig. 2). However, aromatic and heteroaromatic thiols gave much more inferior catalytic results (see **5aj**–**5ao** in Fig. 2).

By using (*R*)-**1d** as the optimal catalyst and benzyl thiol **3a** as the optimal GSH surrogate, we explored the reaction scope with a broad range of aromatic and heteroaromatic 2-oxoaldehydes, as well as alkyl glyoxals. As summarized in Fig. 3, regardless of the electronic and steric nature of the substituents, all aromatic 2-oxoaldehydes **2a**–**2v** reacted with benzyl thiol **3a**, sponta-neously affording the corresponding racemic hemithioacetals, which were then isomerized in the presence of KF, producing the corresponding (*S*)-α-hydroxy thioesters **5a**–**5v** as the final products with excellent yields and particularly high enantio-selectivities. Heteroaromatic oxo-aldehyde **2w** was also smoothly converted into the desired products **5w**, albeit with moderate enantioselectivity. Furthermore, to our delight, highly toxic methylglyoxal **2x** and other primary alkyl glyoxals **2y** and **2z** also underwent the reaction, producing the corresponding (*S*)-α-hydroxy thioesters **5x**–**5z**, respectively, with good to excellent enantioselectivity (88–95% ee).

**Synthetic utility**. Finally, we successfully utilized α-hydroxy-thioesters for the coupling reagent-free synthesis of optically pure α-hydroxyamides, which are very important drug candidates in the pharmaceutical industry[33–38] (Fig. 4a). The reaction of thioester **5f** with aniline **6** in the presence of AgOCOCF₃ (refs 13,39) in THF for 48 h produced hydroxyamide **7** in 87% isolated yield without any loss of enantiopurity. Hydroxyamide **7** is a known intermediate in the synthesis of the selective RAR γ-agonist BMS 270394 (refs 33,40). To our delight, this coupling reaction does not require the use of any coupling reagent and the protection of the OH group. A similar coupling of thioester (*S*)-**5v** with *L*-alanine methylester **8** or lactam alanine amide **10** was successfully used in the synthesis of the γ-secretase inhibitor LY 411575 (refs 34,41) (Fig. 4b, also see Supplementary Methods for experimental details).

## Discussion

To elucidate the reaction mechanism, we carried out the isotope experiments using 1-deuterated-phenylglyoxal. As shown from the catalytic results in Fig. 5, only product **5a** was obtained exclusively. No deuterium incorporation at the α-carbon position of the thioester group in the product clearly indicates that, similar to the real glyoxalase I, the reaction proceeded with deprotona-tion of the α-proton of hemithioacetal with fluoride base to form the enediol intermediate and subsequent enantioselective protonation. It is here noteworthy that the Lewis acid-catalysed

**Table 1 | Optimization of the reaction conditions for the asymmetric isomerization of hemithioacetals.**

(R)-**1a** (X = H), (R)-**1b** (X = Br)
(R)-**1c** (X = I), (R)-**1d** (X = CF$_3$), (R)-**1e** (X = C$_2$F$_5$)

(R)-**1f** (X = I), (R)-**1g** (X = CF$_3$)

(R)-**1h**

(R)-**1i**

| Entry | Cat. | Solvent | Conv. (%) | ee (%) |
|---|---|---|---|---|
| 1 | (R)-**1a** | o-xylene | 65 | 4 |
| 2 | (R)-**1b** | o-xylene | 43 | 56 |
| 3 | (R)-**1c** | o-xylene | 68 | 87 |
| 4 | (R)-**1d** | o-xylene | 99 | 98 |
| 5* | (R)-**1d** | o-xylene | 95 | 95 |
| 6 | (R)-**1e** | o-xylene | 98 | 94 |
| 7[†] | (R)-**1f** | o-xylene | 99 | rac |
| 8[†] | (R)-**1g** | o-xylene | 99 | rac |
| 9 | (R)-**1h** | o-xylene | n.r. | — |
| 10 | (R)-**1i** | o-xylene | n.r. | — |
| 11 | (R)-**1d** | m-xylene | 98 | 97 |
| 12 | (R)-**1d** | p-xylene | 98 | 97 |
| 13 | (R)-**1d** | toluene | 97 | 97 |
| 14 | (R)-**1d** | mesitylene | 86 | 98 |
| 15 | (R)-**1d** | CH$_2$Cl$_2$ | 78 | 91 |
| 16 | (R)-**1d** | THF | 23 | 19 |
| 17 | (R)-**1d** | CH$_3$CN | n.r. | — |

Cat., catalyst; Conv., conversion; n.r., no reaction; rac, racemic.
Unless otherwise indicated, the reactions were performed with **2a** (0.1 mmol, monohydrate form), **3a** (1.0 equiv.), catalyst (10 mol%) and KF (2.0 equiv.) at 20 °C. The conversion was determined by ¹H-NMR analysis of the reaction mixture. The enantiomeric excess was determined by HPLC analysis.
*The reaction was performed with hemithioacetal **4a** (0.1 mmol), catalyst (10 mol%) and KF (2.0 equiv.) at 20 °C.
†Using CsF (2.0 equiv.) instead of KF.

intramolecular Cannizzaro reactions of glyoxals with alcohols were proved to proceed via 1,2-hydride shift mechanism[7,10]. Furthermore, the primary isotope effect on the reaction kinetics was observed, which suggests the deprotonation of the α-proton of hemithioacetal is the rate-determining step (Supplementary Fig. 1). Moreover, an essentially linear relationship between the catalyst and the product ee indicates that the reaction involves only one catalyst in the enantio-determining protonation step, supporting our proposed model shown in Fig. 1b (see Supplementary Table 1).

In summary, we developed an artificial glyoxalase I that successfully catalyses the enantioselective isomerization of the spontaneously formed hemithioacetal adducts between diverse 2-oxoaldehydes and thiols, as GSH surrogates into chiral α-hydroxy thioesters. This reaction is exceptionally enantioselective and the α-hydroxythioester products are of high value for multiple synthetic applications. The applicability was highlighted by the coupling reagent-free synthesis of several optically pure α-hydroxyamides, highly important drug candidates in the pharmaceutical industry. Similar to real enzymes, the enforced proximity of the catalyst and substrates by a chiral cage *in situ* formed by the incorporation of potassium salt can enhance reactivity and efficiently transfer the stereochemical information. Our strategy will provide new scientific insights for developing an artificial enzyme which can outperform the original enzyme. Furthermore, this work would also provide a potential starting point for developing artificial enzymes which might be used for pharmaceutical uses.

## Methods

**Procedure for enantioselective isomerization of hemithioacetals.** In a capped vial, anhydrous *o*-xylene (1.0 ml) was added to a mixture of phenylglyoxal monohydrate **2a** · $H_2O$ (13.4 mg, 0.1 mmol), catalyst (*R*)-**1d** (9.6 mg, 10 mol%) and spray dried KF (11.6 mg, 0.2 mmol, 2.0 equiv.). The reaction mixture was stirred for 1 min at

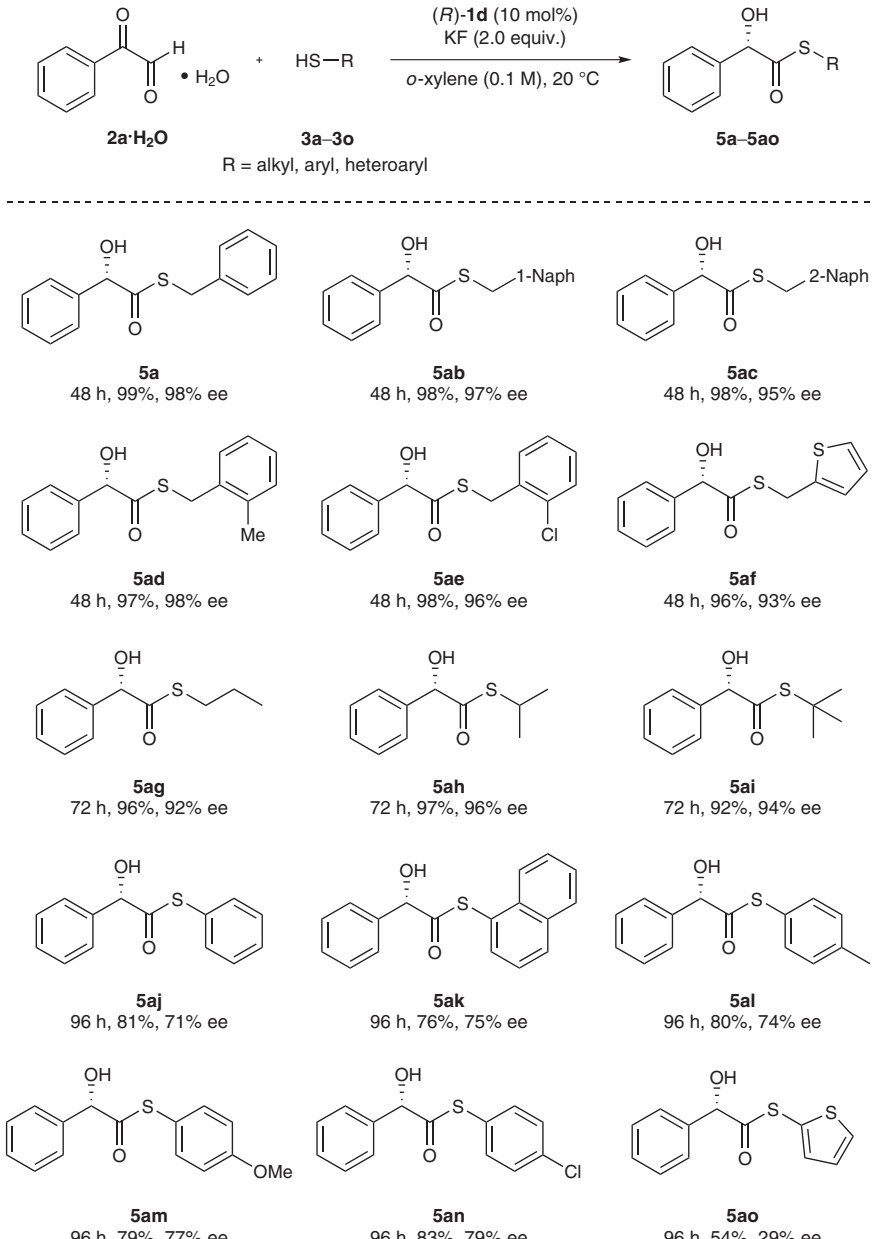

**Figure 2 | Thiol screening.** The reactions were carried out with **2a** (0.1 mmol, monohydrate form), **3** (1.0 equiv.), (*R*)-**1d** (10 mol%) and KF (2.0 equiv.) at 20 °C. Isolated yield.

20 °C. Next, thiol **3a** (12 μl, 0.1 mmol, 1.0 equiv.) was added to the reaction mixture and was further stirred for 48 h at 20 °C. After complete consumption of the hemithioacetal **4a** (following by thin layer chromatography), the mixture was purified using silica gel column chromatography (acetone/*n*-hexane = 1/10) to

obtain the desired product **5a** as colourless liquid in 99% yield and 98% ee. The absolute configuration was determined to be (*S*) by comparison with the optical rotation ([α]$_D$) of the known (*R*)-enantiomer and by analogy the same configuration was assigned to all compounds **5**.

**Figure 3 | Substrate scope of the reaction.** The reactions were carried out with **2** (0.1 mmol, hydrate form), **3a** (1.0 equiv.), (*R*)-**1d** (10 mol%) and KF (2.0 equiv.) at 20 °C. Isolated yield. †Using (*S*)-**1d** (10 mol%). *Using (*R*)-**1d** (5 mol%). **Using (*R*)-**1d** (1 mol%). §Using (*R*)-**1d** (30 mol%) at 0 °C.

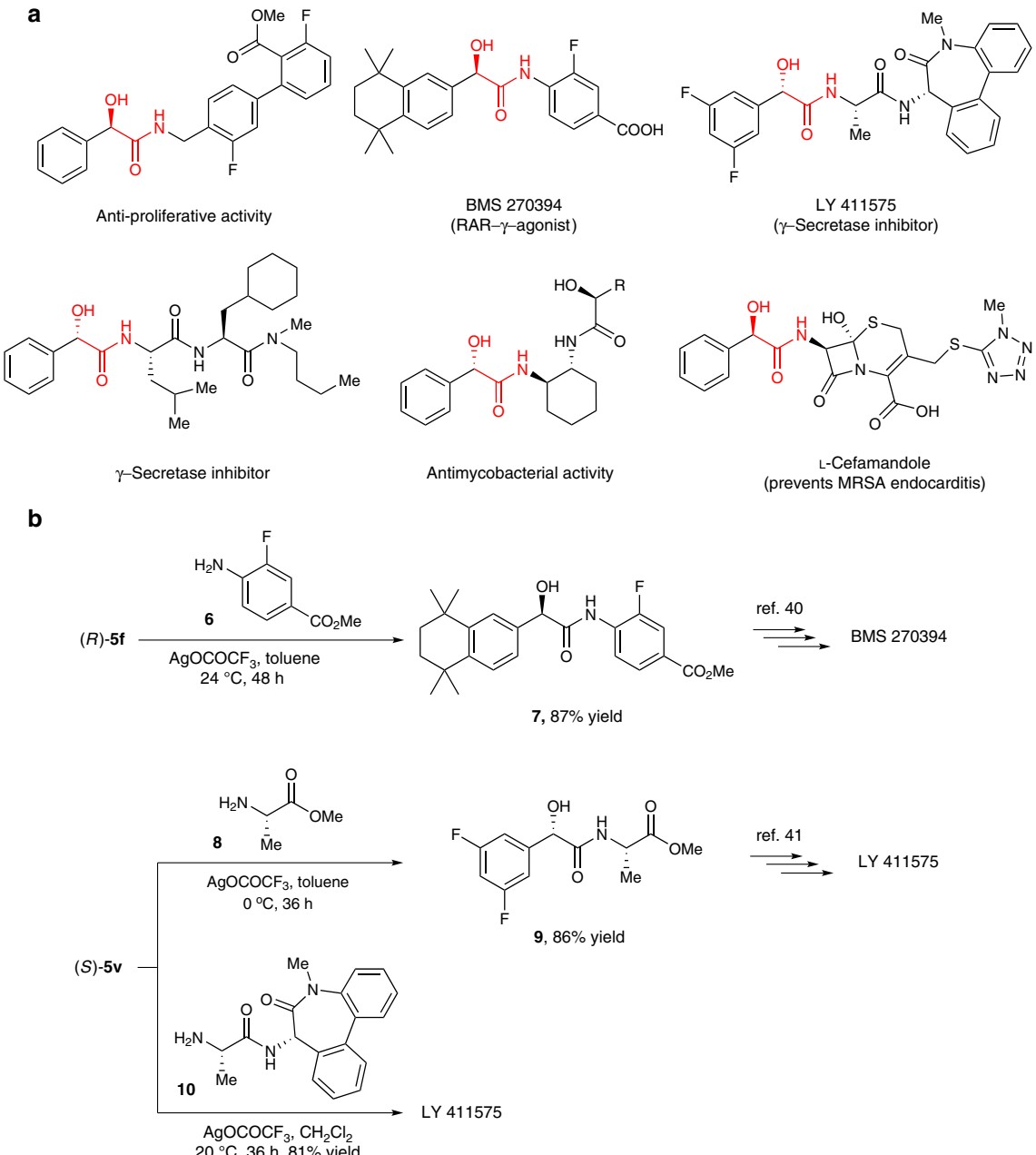

**Figure 4 | Synthetic utility.** (**a**) Pharmaceutically important chiral α-hydroxyamides. (**b**) Synthesis of BMS 270394 and LY 411575 by coupling-reagent-free amidation.

**Figure 5 | Isotope experiment with [1-²H]-phenylglyoxal.** The reaction was performed with [1-²H]-phenylglyoxal (0.1 mmol), **3a** (1.0 equiv.), (*R*)-**1d** (10 mol%) and KF (2.0 equiv.) at 20 °C.

For NMR spectra of the synthesized compounds in this article, see Supplementary Figs 3–46.

For HPLC spectra of **5a–5z**, see Supplementary Figs 47–86. Full experimental details can be found in the Supplementary Methods.

**Data availability.** The authors declare that all relevant data supporting the findings of this study are available within the article and its Supplementary Information files, and also are available from the corresponding author upon reasonable request.

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

## Acknowledgements

We are grateful for the financial support provided by the Ministry of Science, ICT and Future Planning (NRF-2014R1A2A1A01005794 and NRF-2016R1A4A1011451).

## Author contributions

C.E.S. designed and developed the strategy and the mechanistic concepts. S.Y.P., I.-S.H. and H.-J.L. performed the experiments. C.E.S. wrote the manuscript. S.Y.P. prepared the Supplementary Information. All the authors discussed the results and commented on the manuscript.

## Additional information

**Competing interests:** The authors declare no competing financial interests.

**Publisher's note**: 

