## [Peer review file · Nature Communications]

Reviewers' comments:

Reviewer #1 (Remarks to the Author):

The manuscript describes a catalytic system for the formation of highly enantiomerically enriched α -hydroxythioesters by deprotonation of α -oxoaldehydes followed by enantioselective reprotonation. Synthetically, the chemistry is appealing since it provides a straightforward catalytic access to chiral α -hydroxy thioesters, useful intermediates for functional group manipulations and follow-up C-C bond forming reactions. Regarding the conceptual novelty, the catalytic system, which consists of two BINOL derivatives tethered by a PEG chain, was previously developed by the authors and used in other enantioselective transformations (Refs 20-12, 23-24). This is the first time, though, that this catalytic system is applied to promote an enantioselective protonation. This is a difficult achievement, and the fact that the potential of the catalytic system could be expanded to include this transformation is remarkable. In this vein, the difficulties associated with the development of a catalytic enantioselective protonation should be discussed in more details (a useful review has been recently published on the topic, see *Nature Chemistry* 1, 359 - 369 (2009)).

Overall, the work has been carefully conducted, synthetically useful products have been obtained with impressive yields and enantiomeric excesses and the bio-mimetic analogy is interesting. On this basis, the manuscript might be suitable for publication within the journal. However, there are some points to be carefully considered before, as detailed below.

The catalytic system has been developed by the authors few years ago, and it has been successfully applied to promote different stereocontrolled processes. On this basis, it is quite unrealistic for the authors to claim (see the first line in page 4 and in the Conclusion section) that they have developed an artificial enzyme. All along the manuscript the authors stress how difficult it is to develop artificial enzymes ("it is very challenging", to use the authors' own hyperbole), indirectly suggesting that they have succeeded in achieving this goal. But surely the catalyst, which can be ultimately considered to use an enzyme-like mechanism, was not implemented by design and for this purpose, since it has been designed years ago. The text should be amended to tone down this point and other unnecessary hyperbole about the design of artificial enzymes.

Overall, the process closely resembles the intramolecular Cannizzaro reaction, in which α -oxoaldehydes are transformed into the corresponding α -hydroxyesters. The similarities of the two processes should be better recognized. A recently published enantioselective variant of the Cannizzaro reaction can be found here: *J. Am. Chem. Soc.*, 2013, 135 (45), 16849–16852. Importantly, in the Cannizzaro transformation the enantio-determining step is proposed to be the attack of the nucleophile on the glyoxal followed by a stereospecific 1,2-hydride shift. Although this mechanism does not seem to be operational in the reaction catalyzed by glyoxalase I, the possibility of 1,2-hydride shift should be discussed more in depth within the main manuscript. The authors have achieved a strong and direct evidence to exclude the 1,2-hydride shift, but quite surprisingly they have relegated such result in the Supplementary Information. The experiment in the Supplementary Figure 1, which excludes a possible 1,2-hydride shift because of the lack of deuterium incorporation, should be moved into the main text. Along the same line, another important experiment, namely a deuterium experiment suggesting that breaking the C-D bond is the rate determining step, is only discussed in the Supplementary Information.

In page 12, the authors mentioned that the reaction was followed by TLC (monitoring the consumption of compound 2a). This sounds rather odd when considering that the substrate 2a should react spontaneously (and presumably rather fast) with the thiol to form the intermediate 4a (as claimed in the text). Considering that the deprotonation of 4a is rate-limiting, the disappearance of 2a would not assure that completion of the reaction is reached. More information on this point should be given, including more details on the rate of the formation of 4a.

In Figure 1b, the deprotonation of 4a is shown. However, as this is not the stereo-determining step, showing the proposed enediol intermediate being protonated would be more helpful to the reader. In Figure 1a, the nature of the metal involved in the activity of glyoxalase I should be specified (Zn, Mg).

At the end of page 2, it seems that the comments about the decreased activity of glyoxalase I owing to the aging are not pertinent for this study.

Page 3 – “be reprotonated enantiomerically” is incorrect and should read instead: “be stereoselectively reprotonated”.

Reviewer #2 (Remarks to the Author):

The manuscript by Song and coworkers describes their discovery of a biomimetic transformation of oxaldehydes to chiral alpha-hydroxythioesters using an artificial glyoxalase I. The concept of this biomimetic catalyst by mimicking the crown ether for potassium cation binding, resulting in a chiral cage that serves as a base to trigger asymmetric induction is highly important. However, this concept has also been demonstrated for multiple times by the same group and others (Refs 21, 23, 24, etc.). While the present reaction is a new demonstration of this concept with excellent results (yield and enantioselectivity), mechanistically it is not a significant advance. However, the reaction itself is very important and has great potential for application, this work may still be suitable for publication after some revisions.

(1) Please give explanations why the aromatic and heteroaromatic thiols gave much more inferior ee values. How about other aromatic thiols?

(2) Determination of the absolute configuration of compound 5a via coupling-reagent-free amidation gave lower ee value. Why? It seems there is little effect to the chiral center.

(3) In Table 3, only one aliphatic substituent (methyl) was demonstrated. More examples of this type with longer chain should be added.

(4) Reference about non-asymmetric isomerization of hemithioacetal (Tetrahedron Lett. 2867-2868 (1970)) should be cited.

(5) In page 5, Results and Discussion, "sand" should read "and".

(6) How about using nucleophiles other than thiols, such as alcohols and amines?

Point-by-point responses to referees for NCOMMS-16-26683-T

Reviewer #1 (Remarks to the Author):

Comment 1:

The manuscript describes a catalytic system for the formation of highly enantiomerically enriched α -hydroxythioesters by deprotonation of α -oxoaldehydes followed by enantioselective reprotonation. Synthetically, the chemistry is appealing since it provides a straightforward catalytic access to chiral α -hydroxy thioesters, useful intermediates for functional group manipulations and follow-up C-C bond forming reactions. Regarding the conceptual novelty, the catalytic system, which consists of two BINOL derivatives tethered by a PEG chain, was previously developed by the authors and used in other enantioselective transformations (Refs 20-12, 23-24). This is the first time, though, that this catalytic system is applied to promote an enantioselective protonation. This is a difficult achievement, and the fact that the potential of the catalytic system could be expanded to include this transformation is remarkable. In this vein, the difficulties associated with the development of a catalytic enantioselective protonation should be discussed in more details (a useful review has been recently published on the topic, see Nature Chemistry 1, 359 - 369 (2009)). Overall, the work has been carefully conducted, synthetically useful products have been obtained with impressive yields and enantiomeric excesses and the bio-mimetic analogy is interesting. On this basis, the manuscript might be suitable for publication within the journal. However, there are some points to be carefully considered before, as detailed below.

Response:

We appreciate your positive comments. According to the suggestion provided by Reviewer 1, we cited the above-mentioned reference in our revised manuscript (Ref. 3). Furthermore, we also added following sentence in the main text (page 3, lines 4-7). "Of note, in contrast to enzymatic protonation, enantioselective introduction of a proton to transient enediol intermediate via synthetic route might be extremely challenging to control it in terms of enantioselectivity due to the small size of the proton^{3,7-11}."

Comment 2:

The catalytic system has been developed by the authors few years ago, and it has been successfully applied to promote different stereocontrolled processes. On this basis, it is quite unrealistic for the authors to claim (see the first line in page 4 and in the Conclusion section) that they have developed an artificial enzyme. All along the manuscript the authors stress how difficult it is to develop artificial enzymes ("it is very challenging", to use the authors' own hyperbole), indirectly suggesting that they have succeeded in achieving this goal. But surely the catalyst, which can be ultimately considered to use an enzyme-like mechanism, was not implemented by

design and for this purpose, since it has been designed years ago. The text should be amended to tone down this point and other unnecessary hyperbole about the design of artificial enzymes.

Response:

We agree with the suggestion provided by Reviewer 1. Thus, the original sentence “Therefore, developing powerful artificial enzyme is highly interesting from the perspectives of both biology and chemistry..” was changed to the following sentence “Therefore, developing powerful biomimetic catalytic system is highly interesting from the perspectives of both biology and chemistry..”. Moreover, we removed the following sentence. “However, it is a very challenging task.”

Comment 3:

Overall, the process closely resembles the intramolecular Cannizzaro reaction, in which α -oxoaldehydes are transformed into the corresponding α -hydroxyesters. The similarities of the two processes should be better recognized. A recently published enantioselective variant of the Cannizzaro reaction can be found here: J. Am. Chem. Soc., 2013, 135 (45), 16849–16852. Importantly, in the Cannizzaro transformation the enantio-determining step is proposed to be the attack of the nucleophile on the glyoxal followed by a stereospecific 1,2-hydride shift. Although this mechanism does not seem to be operational in the reaction catalyzed by glyoxalase I, the possibility of 1,2-hydride shift should be discussed more in depth within the main manuscript. The authors have achieved a strong and direct evidence to exclude the 1,2-hydride shift, but quite surprisingly they have relegated such result in the Supplementary Information. The experiment in the Supplementary Figure 1, which excludes a possible 1,2-hydride shift because of the lack of deuterium incorporation, should be moved into the main text. Along the same line, another important experiment, namely a deuterium experiment suggesting that breaking the C-D bond is the rate determining step, is only discussed in the Supplementary Information.

Response:

Thank you for your very helpful comments. According to the suggestion provided by Reviewer 1, we discussed our mechanistic study in the main text as follows.

“To elucidate the reaction mechanism, we carried out the isotope experiments using 1-deuterated-phenylglyoxal. As shown from the catalytic results in Fig. 5, only product **5a** was obtained exclusively. No deuterium incorporation at the α -carbon position of the thioester group in the product clearly indicates that, similar to the real glyoxalase I, the reaction proceeded with deprotonation of the α -proton of hemithioacetal with fluoride base to form the enediol intermediate and subsequent reprotonation. It is here noteworthy that the Lewis acid-catalysed intramolecular Cannizzaro reactions of glyoxals with alcohols were proved to proceed via 1,2-hydride shift mechanism^{7,10}. Furthermore, the primary isotope effect on the reaction

kinetics was observed which suggests the deprotonation of the α -proton of hemithioacetal is the rate determining step (Supplementary Fig. 1).”

Comment 4:

In page 12, the authors mentioned that the reaction was followed by TLC (monitoring the consumption of compound 2a). This sounds rather odd when considering that the substrate 2a should react spontaneously (and presumably rather fast) with the thiol to form the intermediate 4a (as claimed in the text). Considering that the deprotonation of 4a is rate-limiting, the disappearance of 2a would not assure that completion of the reaction is reached. More information on this point should be given, including more details on the rate of the formation of 4a.

Response:

Thank you for your kind comment.

“the glyoxal derivative **2a**” was corrected to “the hemithioacetal **4a**”.

Comment 5:

In Figure 1b, the deprotonation of 4a is shown. However, as this is not the stereo-determining step, showing the proposed enediol intermediate being protonated would be more helpful to the reader. In Figure 1a, the nature of the metal involved in the activity of glyoxalase I should be specified (Zn, Mg).

Response:

According to the suggestion provided by Reviewer 1, the Figure 1 was redrawn. In Figure 1a, the nature of the metal involved in the activity of glyoxalase I was also specified (Zn, Ni).

Comment 6:

At the end of page 2, it seems that the comments about the decreased activity of glyoxalase I owing to the aging are not pertinent for this study.

Response:

We agree with the suggestion provided by Reviewer 1. Thus, the following sentences were removed.

“On the other hand, the decreased activity of glyoxalase I owing to the aging process and oxidative stress results in an increase in the toxic α -oxoaldehyde concentration and subsequently causes increased glycation and tissue damage². Thus, the suppression of α -oxoaldehyde-mediated glycation by glyoxalase I is particularly important. On the other hand, glyoxalase I inhibitors, which lead to the accumulation of cytotoxic α -oxoaldehydes, can be used as antitumor and antimalarial agents¹⁶. “

Comment 7:

Page 3 – “be reprotonated enantiomerically” is incorrect and should read instead: “be stereoselectively reprotonated”.

Response:

Thank you for your kind comment.

“be reprotonated enantiomerically” was changed to “be stereoselectively reprotonated”.

Reviewer #2 (Remarks to the Author):**Comment 1:**

The manuscript by Song and coworkers describes their discovery of a biomimetic transformation of oxaldehydes to chiral alpha-hydroxythioesters using an artificial glyoxalase I. The concept of this biomimetic catalyst by mimicking the crown ether for potassium cation binding, resulting in a chiral cage that serves as a base to trigger asymmetric induction is highly important. However, this concept has also been demonstrated for multiple times by the same group and others (Refs 21, 23, 24, etc.). While the present reaction is a new demonstration of this concept with excellent results (yield and enantioselectivity), mechanistically it is not a significant advance. However, the reaction itself is very important and has great potential for application, this work may still be suitable for publication after some revisions.

Response:

We appreciate your highly positive comments.

Comment 2:

(1) Please give explanations why the aromatic and heteroaromatic thiols gave much more inferior ee values. How about other aromatic thiols?

Response:

According to the suggestions provided by Reviewer 2, we examined three more aromatic thiols (**5al**, **5am** and **5an**). The full analytical data of the products were added in the Supporting Information. However, all aromatic and heteroaromatic thiols examined in this study gave much inferior results compared with those obtained with aliphatic thiols. However, at the present stage, we cannot provide the suitable explanations why the aromatic and heteroaromatic thiols gave much more inferior ee values.

Comment 3:

(2) Determination of the absolute configuration of compound **5a** via coupling-reagent-free amidation gave lower ee value. Why? It seems there is little effect to the chiral center.

Response:

A slight racemization was observed in the amidation of α -hydroxy thioesters with benzyl amine in the absence of AgOCOCF_3 at room temperature, due to the acidic α -hydrogen in the thioester product. However, the racemization could be suppressed by employing catalytic amounts of AgOCOCF_3 and lower reaction temperature (-20 °C) (see, Supplementary Information page S111).

Comment 4:

(3) In Table 3, only one aliphatic substituent (methyl) was demonstrated. More examples of this type with longer chain should be added.

Response:

According to the suggestions provided by Reviewer 2, we examined three more aliphatic glyoxals (**2y**, **2z**) to demonstrate the broad scope of our protocol. Good to excellent results were obtained in terms of the yield and enantioselectivity (in Fig. 3: **5y**, 90%, 88% ee; **5z**, 91%, 88% ee). The full analytical data of the products were added in the Supplementary Figs. 41, 42, 85, 86 and Supplementary Information page S125.

Comment 5:

(4) Reference about non-asymmetric isomerization of hemithioacetal (Tetrahedron Lett. 2867-2868 (1970)) should be cited.

Response:

According to the suggestion provided by Reviewer 2, we cited this reference (Ref. 32) in the revised manuscript.

Comment 6:

(5) In page 5, Results and Discussion, "sand" should read "and".

Response:

Thank you for your kind comment.
"sand" was corrected to "and".

Comment 7:

(6) How about using nucleophiles other than thiols, such as alcohols and amines?

Response:

According to the suggestion provided by Reviewer 2, we conducted the reactions of phenyl glyoxal with benzyl alcohol and benzyl amine instead of benzyl thiol. However, the reactions afforded only the corresponding hemiacetal and hemiaminal, respectively. The further conversion of hemiacetal and hemiaminal into the

corresponding α -hydroxyester and α -hydroxy amide, respectively, was not observed, which can be explained by the relatively lower acidity of α -proton of hemiacetals and hemiaminals compared with that of hemithioacetals.